# Improved Algorithms of Data Processing for Dispersive Interferometry Using a Femtosecond Laser

**DOI:** 10.3390/s23104953

**Published:** 2023-05-21

**Authors:** Tao Liu, Jiucheng Wu, Amane Suzuki, Ryo Sato, Hiraku Matsukuma, Wei Gao

**Affiliations:** Precision Nanometrology Laboratory, Department of Finemechanics, Tohoku University, Sendai 980-8579, Japan; liu.tao.q8@dc.tohoku.ac.jp (T.L.); wu.jiucheng.q1@dc.tohoku.ac.jp (J.W.); amane.suzuki.t5@dc.tohoku.ac.jp (A.S.); ryo.sato.t5@dc.tohoku.ac.jp (R.S.); hiraku.matsukuma.d3@tohoku.ac.jp (H.M.)

**Keywords:** absolute distance measurement, dispersive interferometry, Fourier transform, optical frequency comb, excess fraction method

## Abstract

Two algorithms of data processing are proposed to shorten the unmeasurable dead-zone close to the zero-position of measurement, i.e., the minimum working distance of a dispersive interferometer using a femtosecond laser, which is a critical issue in millimeter-order short-range absolute distance measurement. After demonstrating the limitation of the conventional data processing algorithm, the principles of the proposed algorithms, namely the spectral fringe algorithm and the combined algorithm that combines the spectral fringe algorithm with the excess fraction method, are presented, together with simulation results for demonstrating the possibility of the proposed algorithms for shortening the dead-zone with high accuracy. An experimental setup of a dispersive interferometer is also constructed for implementing the proposed data processing algorithms over spectral interference signals. Experimental results demonstrate that the dead-zone using the proposed algorithms can be as small as half of that of the conventional algorithm while measurement accuracy can be further improved using the combined algorithm.

## 1. Introduction

Absolute distance measurement with high resolution is important in the field of dimensional metrology. The use of light waves as a length ruler, which was proposed by Michelson in 1893, revolutionized distance measurement by determining the phase shift of a signal after it traveled a certain distance with a subwavelength resolution. However, this method has its limitations, as it can only determine relative distance changes and cannot directly measure the absolute value of the target distance [1,2,3]. Moreover, the phase shift information repeats every fringe, making it difficult to implement measurements beyond λ/2 because the unknown integer fringe order cannot be determined using the Michelson interferometer itself [4,5]. Since its invention at the end of the 20th century [6,7,8], the optical frequency comb (OFC) has been widely employed to broaden the horizon for optical metrology, including measurement of optical frequency [9,10,11], distance [12,13,14,15,16], angle [17,18,19,20,21,22], etc. An optical frequency comb (OFC), which provides numerous ultra-narrow linewidth wavelengths over a broad optical spectral range, is the Fourier transform of femtosecond ultrashort pulses from a mode-locked laser in the time-domain. It enables versatile advanced absolute distance measurement methods, including synthetic wavelength interferometry (SWI) [12,23,24,25], multi-wavelength interferometry (MWI) [26,27,28,29,30], dispersive interferometry (also known as spectrally resolved interferometry (SRI)) [31,32,33,34], dual-comb interferometry [35,36,37,38,39], and time-of-flight (TOF) measurement [40,41,42,43,44]. Compared to other methods, dispersive interferometry can achieve a high resolution of distance measurement.

In 2006, Joo et al. first proposed using the dispersive interference of a mode-locked femtosecond laser to achieve an absolute distance measurement over a range of 0.89 m with a resolution of 7 nm, in which the target distance was obtained by the first-order derivation of the unwrapped phase, while the existence of the direct-current item in the transferred time-domain data generated a certain range of unmeasurable dead-zone close to the zero-position of measurements, i.e., the minimum working distance, in this configuration [14,45]. In 2012, van den Berg et al. made a refinement by combining dispersive and homodyne interferometry based on a virtually imaged phase array (VIPA) and a grating in an optical setup, achieving a non-ambiguous range of 15 cm with an accuracy of *λ*/30, in which the homodyne interferometry is just an auxiliary and cannot output the target distance independently, and the phase is obtained using the cosine fit rather than the wrapped phase [46]. In 2019, Tang et al. proposed a non-filtered and differential envelope phase demodulation method based on dispersive interferometry, where the phase value was obtained by taking the arccosine of a demodulated spectral interference signal [47]. However, the measurement accuracy was influenced by that of the demodulation process for removing the upper and lower envelopes of the spectral interference signal. In addition, the focus of the previous research was on long absolute distance measurement, rather than on shortening the unmeasurable dead-zone close to the zero-position of measurement, which is a critical issue for short-range absolute distance measurement.

In this paper, after a demonstration of the mechanism for generating the dead-zone in the conventional data processing algorithm for dispersive interferometry, two new data processing algorithms are proposed to shorten the dead-zone for millimeter-order short-range absolute distance measurement. The first one is named the spectral fringe algorithm, which is then combined with the excess fraction method [48,49] as the second algorithm. The latter is named the combined algorithm for clarity. The feasibility of the proposed algorithms is then confirmed using simulation and experiment through comparison with the results of the conventional algorithm.

## 2. Principles of the Data Processing Algorithms

### 2.1. Proposed Algorithm 1: The Spectral Fringe Algorithm

The dispersive interferometry using an OFC is typically implemented using a Michelson interferometer-type configuration, whereby the optical path difference (OPD) between the reference and measurement arms can be accurately determined by analyzing the interference spectrum with a proper data processing algorithm [45]. In this subsection, the conventional data processing algorithm is revisited according to the literature [47,50,51], based on which an improved data processing algorithm, namely the spectral fringe algorithm, is proposed.

When a laser beam is emitted from an OFC source, it will be separated by a beam splitter into two beams, which are reflected using mirrors in the reference arm and the measurement arm, respectively. The recombined beams interfere with each other to produce a spectral interference signal, which is subsequently detected using an optical spectrum analyzer (OSA). An OSA has a set of discrete spectral output data. For simplicity, the *k*th spectral output of an OSA is assumed to correspond to the frequency *f_k_* and the wavelength *λ_k_*. Practically, it is difficult to ensure that the splitting ratio (*α_R_*, *α_M_*) of the beam splitter and the reflective index (*r_R_*, *r_M_*) of the mirrors in the two arms are exactly equal. Therefore, the electric fields of the reference beam $E_{R}^{k}\left(t\right)$ and the measurement beam $E_{M}^{k}\left(t\right)$, which correspond to the *k*th spectral output of the OSA, i.e., the *k*th mode of the optical frequency comb if the OSA has enough resolving power, can be expressed as:(1)$$E_{R}^{k}\left(t\right)=\alpha_{R}\cdot r_{R}\cdot E_{0}\cdot G^{\prime }(f_{k})\cdot e^{j\cdot 2\pi \cdot f_{k}\cdot t}$$
(2)$$E_{M}^{k}\left(t\right)=\alpha_{M}\cdot r_{M}\cdot E_{0}\cdot G^{\prime }(f_{k})\cdot e^{j\cdot 2\pi \cdot f_{k}\cdot \left(t-\tau \right)}$$

Here, *τ* is the time delay caused by the optical path difference 2*n*(*f*)*L* between the reference beam and the measurement beam, which is explicitly given as *τ =* 2*n*(*f*)*L/c* where *n* and *c* are the refractive index of air and light speed in a vacuum, respectively. The intensity of the corresponding interference signal can be written as:(3)$$\begin{array}{c} I^{k}\left(t\right)=\frac{1}{2}\cdot \langle E^{k}\left(t\right)\cdot E^{k*}{(t)\rangle }_{T}=\frac{1}{2}\cdot \langle \left[E_{R}^{k}{(t)+E}_{M}^{k}\left(t\right)\right]\cdot {\left[E_{R}^{k*}{(t)+E}_{M}^{k*}\left(t\right)\right]\rangle }_{T}\\ =\frac{1}{2}\cdot \langle E_{R}^{k}\left(t\right)\cdot E_{R}^{k*}{\left(t\right)\rangle }_{T}+\frac{1}{2}\cdot \langle E_{M}^{k}\left(t\right)\cdot E_{M}^{k*}{\left(t\right)\rangle }_{T}+\frac{1}{2}\cdot \langle E_{R}^{k}\left(t\right)\cdot E_{M}^{k*}{\left(t\right)\rangle }_{T}+\frac{1}{2}\cdot \langle E_{R}^{k*}\left(t\right)\cdot E_{M}^{k}{\left(t\right)\rangle }_{T}\\ =E_{0}^{2}\cdot G^{\prime 2}{(f}_{k})\cdot \left[\frac{1}{2}\cdot \alpha_{R}^{2}\cdot r_{R}^{2}+\frac{1}{2}\cdot \alpha_{M}^{2}\cdot r_{M}^{2}{+\alpha }_{R}\cdot r_{R}\cdot \alpha_{M}\cdot r_{M}\cdot \cos\left({2\pi f}_{k}\tau \right)\right]\\ =\frac{1}{2}\cdot {(\alpha }_{R}^{2}\cdot r_{R}^{2}{+\alpha }_{M}^{2}\cdot r_{M}^{2})\cdot E_{0}^{2}\cdot G^{\prime 2}{(f}_{k})\cdot \left[1+\frac{2\cdot \alpha_{R}\cdot r_{R}\cdot \alpha_{M}\cdot r_{M}}{\alpha_{R}^{2}\cdot r_{R}^{2}{+\alpha }_{M}^{2}\cdot r_{M}^{2}}\cdot \cos\left({2\pi f}_{k}\tau \right)\right] \end{array}$$
where $E_{R}^{k*}\left(t\right)$ and $E_{M}^{k*}\left(t\right)$ are the complex conjugates of the electric fields in the reference and measurement beam, respectively. For simplicity, *S*(*f_k_*), which corresponds to the power spectrum of the OFC source, is referred to as the envelope component. The term in the brackets includes a 1 and a cosine function, which is referred to as the interference fringe component with its unwrapped phase and amplitude defined by Φ(*f_k_*) and *A*, respectively. *S*(*f_k_*), Φ(*f_k_*), and *A* can be written as follows:(4)$$S(f_{k})=\frac{1}{2}\cdot {(\alpha }_{R}^{2}\cdot r_{R}^{2}{+\alpha }_{M}^{2}\cdot r_{M}^{2})\cdot E_{0}^{2}\cdot G^{\prime 2}{(f}_{k})$$
(5)$$\Phi (f_{k})={2\pi f}_{k}\tau =2\pi \cdot n+\varphi (f_{k})$$
(6)$$A=\frac{2\cdot \alpha_{R}\cdot r_{R}\cdot \alpha_{M}\cdot r_{M}}{\alpha_{R}^{2}\cdot r_{R}^{2}{+\alpha }_{M}^{2}\cdot r_{M}^{2}}$$
where *φ*($f_{k}$) is the wrapped phase, and *n* is an integer showing the period number of the spectral interference fringe component. Consequently, the intensity of the spectral interference signal output from the OSA, which is a Fourier transform of Equation (3), can be rewritten as:(7)$$I(f_{k})=S(f_{k})\cdot \left[1+A\cdot \cos\left({2\pi f}_{k}\tau \right)\right]$$

In the conventional data processing algorithm, the spectral interference signal shown in Equation (7) is a directly inverse Fourier transformed into a time-domain function *i*(*t*) as follows:(8)$$i(t)=s(t)⊗\left[\delta (t)+\frac{1}{2}\delta (t-\tau )+\frac{1}{2}\delta (t+\tau )\right]=s(t)+\frac{A}{2}\cdot S(t-\tau )+\frac{A}{2}\cdot S(t+\tau )$$
where δ(t) is a unit impulse function, and *s*(*t*) is the inverse Fourier transform of *S*(*f*). Assuming the OFC source has a Gaussian-like power spectrum, both *S*(*f*) and *s*(*t*) will have Gaussian-like shapes. Three Gaussian-like pulses can then be observed in *i*(*t*), with their peaks located at −*τ*, 0, and *τ* and amplitudes of $\frac{A}{2}$∙$\left|S(t-\tau )\right|$, *A*∙$\left|S(t)\right|$, and $\frac{A}{2}$∙$\left|S(t+\tau )\right|$, respectively. The pulse at *τ* is subsequently picked out by utilizing a time-window centered at *τ*. Then, the picked-up pulse is Fourier transformed into the frequency domain as follows:(9)$$I^{\prime }\left(f\right)=\frac{A}{2}\cdot S(f)\cdot e^{-j2\pi f\tau }$$

The wrapped phase *φ*(*f*) can then be calculated using the arctangent function as:(10)φ(f)=tan−1−ImI′fReI′f 

Since the wrapped phase *φ*(*f*) changes periodically within the range of [−π/2, +π/2] with the increase or decrease of the target distance *L*, *L* cannot be obtained directly from *φ*(*f*) due to the ambiguity in period number *n* in Equation (5). Therefore, it is necessary to calculate *L* by taking the first-order derivation of the wrapped or the unwrapped phase values as follows:(11)$$\frac{d\Phi (f)}{df}=\frac{d\varphi (f)}{df}=\frac{{4\pi n}_{g}L}{c}$$
where $n_{g}$ = *n*(*f*) + (*dn*(*f*)/*df*)∙*f* is the group refractivity of air determined by the central frequency of the source, and *n*(*f*) is the phase refractivity of air. *L* can then be obtained using:(12)$$L=\frac{c}{{4\pi n}_{g}}\cdot \frac{d\Phi (f)}{df}=\frac{c}{{4\pi n}_{g}}\cdot \frac{d\varphi (f)}{df}$$

As can be seen above, in the conventional data processing algorithm for dispersive interferometry, the spectral interference signal from the OSA shown in Equation (7) is directly inverse Fourier transformed into the time-domain function *i*(*t*) shown in Equation (8). In this conventional algorithm, the dead-zone, which is the unmeasurable distance range, is determined by the width of the pulse function *s*(*t*). Since τ is proportional to the target distance *L*, when *L* is smaller than a certain value, the three pulses of *i*(*t*) will overlap with each other. In this case, the time-window for selecting the pulse does not function well, which prevents the measurement of the distance.

On the other hand, in the proposed spectral fringe algorithm, the envelope component *S*(*f*) and the 1 in the brackets of Equation (7) are removed to leave only the cosine function, i.e., the interference fringe component as a modified spectral interference signal *I_m_*(*f*) = $A\cdot \cos\left(2\pi f\tau \right)$. Consequently, the inverse Fourier transform of the modified *I_m_*(*f*) will generate a modified time function of *i_m_*(*t*) with only two impulse-shaped pulses at −*τ* and *τ*, instead of the three Gaussian-like pulses at −*τ*, 0, and *τ* in the conventional data processing algorithm. As a result, the distance between two pulses in *i_m_*(*t*) using the spectral fringe algorithm is doubled due to the removal of the central pulse compared with that in *i*(*t*) using the conventional algorithm. In addition, the width of the impulse-shaped pulses *i_m_*(*t*) is much narrower than that of the Gaussian-like pulses in *i*(*t*). These two effects can significantly shorten the dead-zone, which is the fundamental concept of the proposed spectral fringe algorithm shown below.

In the spectral fringe algorithm, the upper and lower envelopes of the detected spectral interference signal in Equation (7) are first evaluated based on [47] as follows:*UE*(*f_k_*) = *S*(*f_k_*)∙(1 + *A*)(13)
*LE*(*f_k_*) = *S*(*f_k_*)∙(1 − *A*)(14)

The modified spectral interference signal *I_m_*($f_{k}$) can then be obtained using:(15)$$I_{m}(f_{k})=A\cdot \cos\left({2\pi f}_{k}\tau \right)=\frac{I\left(f_{k}\right)}{{S(f}_{k})}-1$$
where
(16)$$S(f_{k})=\frac{1}{2}\cdot [\mathrm{UE}(f_{k})+\mathrm{LE}(f_{k})]$$
(17)$$A=\frac{{\mathrm{UE}(f}_{k})-{\mathrm{LE}(f}_{k})}{{\mathrm{UE}(f}_{k}{)+\mathrm{LE}(f}_{k})}$$

Subsequently, an inverse Fourier transforming of Equation (15) gives rise to modified time function *i_m_*(*t*) as follows:(18)$$i_{m}(t)={\mathrm{FT}}^{\;-1}\left[{I_{m}(f}_{k})\right]=\frac{A}{2}\cdot \delta (t+\tau )+\frac{A}{2}\cdot \delta (t-\tau )$$

Differing from the conventional data processing algorithm, only two impulse-shaped pulses exist in the time-domain; they are located at −*τ* and *τ* with an intensity of $\frac{A}{2}$∙δ(t+τ) and $\frac{A}{2}$∙δ(t−τ), respectively. The distance between the two pulses is twice of that in the conventional algorithm, which shortens the dead-zone to be half for the same pulse width. In addition, the width of the impulse-shaped pulses using the proposed spectral fringe algorithm is much narrower than that of the Gaussian-like pulses using the conventional algorithm, which further shortens the dead-zone. In other words, the impulse-shaped pulse at *τ* can be selected more easily and precisely using a time-window with a much narrower bandwidth for calculating the target distance based on Equations (9)–(12) with a significantly shortened dead-zone.

It should be noted that although the methods of calculating the upper and lower envelopes of the spectral interference signal are similar, the proposed spectral fringe algorithm is different from that of [47] from the point of view of the phase calculation method. In [47], the phase of the spectral interference signal is directly obtained from taking arccosine of the modified spectral interference signal *I_m_*(*f*) based on Equation (15) for the purpose of improving the performance of dispersive interferometry in long absolute distance measurement. On the other hand, in the proposed spectral fringe algorithm, the phase is obtained from Equations (9) and (10) by using the time-windowed impulse-shaped pulse of the modified time function *i_m_*(*t*) in Equation (18) for the purpose of shortening the dead-zone close to the zero-position of measurement, which is a critical issue in millimeter-order short-range absolute distance measurement.

### 2.2. Proposed Algorithm 2: The Combined Algorithm

The spectral fringe algorithm is combined with the excess fraction method [52] as the combined algorithm. In a dispersive interferometer using an OFC, the relationship between the *k*th OSA output of wavelength *λ_k_* (*λ_k_ = c*/(2π$f_{k}$)) and the target distance *L* can be given by *n_i_*∙*L* = (*m_k_* + *ε_k_*)∙*λ_k_*, in which *m_k_* is an integer, *ε_k_* is the excess fraction part of the wavelength *λ_k_*, and *n_i_* is the refractive index. The excess fraction part *ε_k_* can be calculated using the wrapped phase *φ*(*f_k_*), which is *ε_k_* = [*φ*(*f_k_*) + 0.5π]/π. The uncertain integer number *m_k_* can be obtained from the excess fraction part *ε_k_* based on the excess fraction method.

The excess fraction method is an effective approach to determine the absolute target length *L* by employing the measured excess fraction values of multiple wavelengths *λ*_1_ > *λ*_2_ > … > *λ_i_* [53]. For the dispersive interferometer with an OFC, multiple discrete spectral outputs of the OSA with a narrow linewidth determined by the wavelength resolution of that OSA, which can be as small as 0.02 nm [54], can be employed for the excess fraction method. The relationship between the target distance *L* and the used wavelengths can be expressed as:(19)$$L=(N_{1\mathrm{real}}+\varepsilon_{1})\cdot \frac{\lambda_{1}}{2}=(N_{2\mathrm{real}}+\varepsilon_{2})\cdot \frac{\lambda_{2}}{2}=\ldots =(N_{\mathrm{ireal}}+\varepsilon_{i})\cdot \frac{\lambda_{i}}{2}$$
where *N_ireal_* is the unknown integer part of the wavelength order for each wavelength *λ_i_*, *ε_i_* is the measured fractional fringe value whose value is within the interval of [0, 1], and the subscript *i* is the wavelength number. Similarly, the nominal length $L^{\prime }$ measured using another method with an error range of ${\Delta L}_{\mathrm{err}}$ can be expressed as:(20)$$L^{\prime }=(N_{1}^{\prime }+\varepsilon_{1}^{\prime })\cdot \frac{\lambda_{1}}{2}=(N_{2}^{\prime }+\varepsilon_{2}^{\prime })\cdot \frac{\lambda_{2}}{2}=\ldots =(N_{i}^{\prime }+\varepsilon_{i}^{\prime })\cdot \frac{\lambda_{i}}{2}$$

Here, $N_{i}^{\prime }$ and $\varepsilon_{i}^{\prime }$ represent the calculated integer part and the excess fraction part of the wavelength order, respectively. $\varepsilon_{i}^{\prime }$ can be calculated using:(21)$$\varepsilon_{i}^{\prime }=\frac{{2L}^{\prime }}{\lambda_{i}}-\mathrm{INT}(\frac{{2L}^{\prime }}{\lambda_{i}})$$
in which function *INT*(*x*) represents obtaining the integer part value of *x*. The difference *c* between the real length *L* and the nominal length $L^{\prime }$ is:
(22)$$c=L-L^{\prime }=(m_{i}+\delta_{i})\cdot \frac{\lambda_{i}}{2}=m_{i}\cdot \frac{\lambda_{i}}{2}+\delta_{i}\cdot \frac{\lambda_{i}}{2}$$
where the first term *m_i_*$\cdot \frac{\lambda_{i}}{2}$ = (*N_ireal_* − $N_{i}^{\prime }$)$\cdot \frac{\lambda_{i}}{2}$ represents the integer part of the difference *c*, and its value is varied by adjusting the desired value of $N_{i}^{\prime }$. The second term *δ_i_*$\cdot \frac{\lambda_{i}}{2}=$ (*ε_i_* − $\varepsilon_{i}^{\prime }$)$\cdot \frac{\lambda_{i}}{2}$ is stable and represents the excess fractional part of *c*.

Now, the issue of determining the integer part of the wavelength order *N_ireal_* is transformed to determining the value of *c*. The maximum adjustment value ∆*N_iT_* for the calculated integer part $N_{i}^{\prime }$ is:(23)$$∆N_{\mathrm{iT}}=\mathrm{INT}(\frac{{2\Delta L}_{\mathrm{err}}}{\lambda_{i}})$$
where ${\Delta L}_{\mathrm{err}}$ represents the error range of the nominal length. Hence, the acceptable adjustment integer value *m_ij_* for the integer part of *c* is within the interval:
−∆*N_iT_* ≤ *m*_ij_ = *j* ≤ ∆*N_iT_*(24)

As a result, the value for difference *c* can be adjusted using a variable *m_ij_*,
(25)$$c_{j}=(m_{\mathrm{ij}}+\delta_{i})\cdot \frac{\lambda_{i}}{2}$$

Considering the same adjusted difference value of *c_j_* when using another wavelength *λ_i_*_+1_, the fractional part of wavelength order caused by *c_j_* can be calculated using:(26)$$\delta_{(i+1)j}=\frac{{2c}_{j}}{\lambda_{i+1}}-\mathrm{INT}(\frac{{2c}_{j}}{\lambda_{i+1}})$$

Hence, the fractional part of $L^{\prime }\;$+ *c_j_* using the wavelength *λ_i_*_+1_ can be given by:(27)$$\varepsilon_{(i+1)j}=\delta_{(i+1)j}+\varepsilon_{\left(i+1\right)}^{\prime }$$

Finally, when the difference value between *ε*_(*i*+1)*j*_ and *ε*_(*i*+1)_ reach the minimum, the corresponding *c_J_* is the optimum adjusted value, and the real length *L* of the target distance can be calculated using:(28)$$L=L^{\prime }+c_{J}$$

Figure 1 illustrates the detailed procedure of selecting the optimum difference *c_J_* and the relationship between those above-mentioned parameters in the excess fraction method.

It is worth mentioning that similar to [46], the target distance *L* is given by *n_i_*∙*L* = (*m_k_* + *ε_k_*)∙*λ_k_*. However, the approach in the proposed combined algorithm for determining the integer and excess fraction part of wavelength *λ_k_* is different from that of [46]. In [46], the phase *φ*(*λ_k_*) for a specific wavelength *λ_k_* is obtained from a cosine fit of the interference signal, and the integer number of the wavelength *m_k_* is determined by the result of dispersive interferometry. In addition, the aim of using the integer and excess fraction part of wavelength *λ_k_* in [46] is not to measure the absolute distance *L* directly and make a comparison with the result of the dispersive interferometry, but to refine the final value of the distance *L* by averaging the results of different wavelengths.

## 3. Simulation and Experiment Results

### 3.1. Simulation Results

Simulation results of the interference spectrum detected using the OSA with different target distances of 1 mm, 2 mm, and 3 mm without considering the inequality of laser power in the two arms are shown in Figure 2. In the simulation process, parameters of the femtosecond laser source are set as center frequency *f_c_* = 192.175 THz, repetition frequency *f_rep_* = 100 MHz, and full width as half maximum FWHM = 2.5 THz.

It can be seen from Figure 2 that with increasing the target distance from 1 mm to 3 mm, the density of the interference fringes dramatically raises. If the target distance continuously stretches to a certain threshold, the interference fringes will become too dense to be distinguished due to the limited resolution of the OSA. To expand the applicable range for dispersive interferometry, decreasing the fringes’ density at large distances is the key point, which can be performed using several approaches. For instance, employing a Fabry-Perot Etalon (FPE) before the spectrometer to trim down the incident combs’ density is one approach, or one can just choose a femtosecond source with a higher repetition frequency, which has a larger comb interval.

In an actual experiment, it is impossible to ensure that the optical power in the reference and the measurement arms of a Michelson-type interferometer are exactly equal. Therefore, it is critical to clarify the impact derived from modulated parameter *A*, which reflects the inequality of the optical power in the two arms. For a certain target distance of *L* = 1 mm, simulation results of different values of modulated parameter *A* = 0.25, 0.5, and 0.75 are shown in Figure 3.

Compared to Figure 2a where the modulated parameter *A* is set to 1, i.e., the optical power in the two arms is equal, a decreasing value of *A* leads to an increasing inflation of the lower envelope away from the zero line and a significant reduction in the visibility *V* of the interference fringes.

#### 3.1.1. Simulation Results of the Conventional Data Processing Algorithm

Based on the interference spectrum with a target distance of 1 mm shown in Figure 2a, the data processing procedure of the conventional algorithm is shown in Figure 4. Inverse Fourier transforming, time-window selecting, and phase unwrapping processes are operated to generate the desired distance using Equation (12) with a result of 1.0005 mm. The deviation of 0.5 μm between the simulated result and the ideal target distance may be caused by the error or signal loss in the time-window selecting and phase unwrapping; this is verified by other researchers [55]. It is worth mentioning that for the inverse Fourier transforming in Figure 4a, the target distance can be evaluated coarsely using the time peak position, 2*L* = *c*∙*X*(*t* = *τ*), and the modulated parameter *A* can be assessed coarsely using the ratio of *A* = *Y*(*t* = 0)/2*Y*(*t* = *τ*).

#### 3.1.2. Simulation Results of the Spectral Fringe Algorithm

An interference spectrum with a target distance of 1 mm and a modulated parameter *A* = 0.5 is employed in the simulated data processing procedure of the spectral fringe algorithm. Firstly, it extracts the upper and lower envelopes to obtain the modified spectral interference signal *I_m_*($f_{k}$), and then it operates inverse Fourier transforming to achieve two sharp peaks in the time-domain. Subsequently, it obtains the wrapped phase from the Fourier transform results of the time-window filtered peak, and finally, the target distance is developed using the first-order deviation of the unwrapped phase, as illustrated in Figure 5.

To clarify the advantage of the proposed spectral fringe algorithm, a simulation is carried out to compare it with the conventional data processing algorithm for measurement of a short target distance *L* of 100 μm. As shown in Figure 6a, there are three time pulses in the inverse Fourier transform results for the spectral interference signal using the conventional algorithm, located at −*τ*, 0, and *τ*, respectively. Since the pulses have a large pulse width, they overlap with each other for the target distance *L* of 100 μm, which makes it difficult to select the pulse at *τ* using a time-window with high accuracy. Consequently, a large error of distance occurs in the subsequent data processing for distance calculation. In contrast, as shown in Figure 6b, for the proposed spectral fringe algorithm, there are only two impulse-shaped pulses located at −*τ* and *τ*, respectively, with a very narrow pulse width and without the existence of the central pulse. For the target distance *L* of 100 μm, the two impulse-shaped pulses are separated completely owing to the narrow pulse width and the elimination of the central pulse. This makes it possible to accurately select the pulse at *τ* using a time-window, which significantly increases the distance measurement accuracy using the spectral fringe algorithm. Based on the simulation results, the target distance of 100 μm is measured to be 96.005 μm and 99.824 μm using the conventional data processing algorithm and the proposed spectral fringe algorithm, respectively. The corresponding distance measurement errors are 3.99 μm and 0.18 μm, respectively, which demonstrates the feasibility of the proposed spectral fringe algorithm for improving the accuracy of distance measurement.

#### 3.1.3. Simulation Results of the Combined Algorithm

The simulated data processing procedure of the combined algorithm is started by obtaining the wrapped phase *φ*(*f_k_*) in the spectral fringe algorithm, as shown in Figure 7. The excess fraction *ε_k_* is then calculated using *ε_k_* = [*φ*(*f_k_*) + 0.5π]/π. Two arbitrary points of excess fraction value in Figure 7b, Point A (190.519 THz, 0.253494) and Point B (191.988 THz, 0.845917), are selected to achieve the desired target length based on the excess fraction method using Equation (28) with a result of 0.99995 mm. The deviation between the simulated result and the ideal target distance decreases to 0.05 μm. Furthermore, it is also possible to select several excess fraction values within the OSA working range to calculate the target distances individually and average their results to improve the accuracy and stability of the final result.

### 3.2. Experimental Setup

An overview of the experimental setup is shown in Figure 8. A homemade mode-locked femtosecond fiber laser source with a center frequency *f_c_* of 192.175 THz is exploited as the frequency comb for the experiment [56]. The time width of a single pulse is ~142 fs, the repetition frequency *f_rep_* is 100 MHz, and its spectral FWHM is ~62.4 nm with an energy of ~5 × 10^−11^ J per pulse. About 5 mW power from a laser is sent into the beam splitter (OptoSigma, NPCH-10-15500, Santa Ana, CA, USA) via a single-mode fiber. Two square-protected silver mirrors (Thorlabs, PFSQ05-03-P01, Newton, NJ, USA) are utilized to reflect the light in the reference and measurement beam, separately. The mirror in the measurement beam is mounted on a single-axis motorized stage (Saruka Seiki, KXC04015-CA, Osaka, Japan), so that it can be moved linearly in the optical path direction within a travel range of ±7 mm. It is worth noting that when the measurement and reference arms were equal, the motorized stage was in the vicinity of −3.8 mm, giving rise to the maximum available measurement distance of 10 mm in the experimental setup. Furthermore, a laser encoder (Renishaw, RLD10-3R, Wotton-under-Edge, UK) is employed as an external reference to calibrate the moving distance of the stage. The interference optical signal is subsequently analyzed using an optical spectrum analyzer (Yokogawa, AQ6370D, Incheon, Republic of Korea) with a wavelength resolution of 0.02 nm. Moreover, it took approximately 1 s to collect and save the OSA data as well as to move the motorized stage to the next sampling point. The sampling rate is thus approximately 1 Hz in the experimental setup.

### 3.3. Experimental Results and Discussion

In the experiment, the target mirror in the measurement arm was moved continuously from −1 mm to +10 mm with a step of 10 μm, and the interference signal within the frequency range of 191 THz to 193 THz was recorded using the OSA at the same time. The laser encoder was set to zero at the initial position to calibrate the movement distance of the motorized stage. The experimental interference spectrum at various positions is shown in Figure 9, in which it is easy to see that the interference fringe density increases, and the visibility of the interference signal decreases due to the effect of environmental disturbance and air divergence with increasing length of the measurement arm, complying with the simulation results from Section 3.1.

Furthermore, the detected interference spectrum at different positions was processed using the conventional data processing algorithm, the spectral fringe algorithm, and the combined algorithm, respectively. Comparisons of the measurement results of these three algorithms are presented in Figure 10, Figure 11, Figure 12 and Figure 13.

Both the conventional data processing algorithm and the spectral fringe algorithm revealed a good agreement with the reference distance for a target distance from +1 mm to +10 mm, as illustrated in Figure 10. There was little manifest difference in the measurement results of these two algorithms when the target distance was not a short one because the pulse interval in the time-domain was increased compared to that of a short target distance, making the time pulse easy to be picked up by a time-window. Meanwhile, for a target distance from +1 mm to +10 mm, the measurement results of the combined algorithm were also highly consistent with the reference distance, as shown in Figure 11a. However, Figure 11b illustrates that the difference between the measurement results and referenced distance of the combined algorithm is nearly two times smaller than those of the conventional and the spectral fringe ones, which are ~1.8 μm and ~4.2 μm, respectively. In other words, the combined algorithm can improve the accuracy of the conventional data processing algorithm owing to utilizing the wrapped phase directly.

The performance of the conventional data processing algorithm dramatically decreased when the target distance was smaller than 1 mm. The measurement results of the conventional data processing algorithm from −1 to +1 mm are shown in Figure 12, in which an undetectable region, i.e., the dead-zone, emerged near the zero distance. The undetectable region is highlighted in a red box, whose amplified details are shown in Figure 12b. The dead-zone of the conventional data processing algorithm was approximately 200 μm, where the target distance could not be measured due to the overlap of time-domain pulses in the inverse Fourier transform results of the spectral interference signal.

However, the two proposed algorithms could reach more precise results within the dead-zone of the conventional algorithm, as shown in Figure 13. The yellow region in Figure 13a reflects the measurement results derived from these three algorithms with a target distance from 100 μm to 200 μm, in which the two proposed algorithms showed a good agreement with the reference distance. The minimum working distance achieved by use of the proposed spectral fringe algorithm was approximately 100 μm, while that of the conventional one was around 200 μm, which indicated the spectral fringe algorithm could significantly shorten the dead-zone to be two times smaller than that of the conventional one. This is due to the vanished central pulse and the sharper pulse shape in the time-domain of the inverse Fourier transform results of the spectral interference signal, as shown in Figure 13b. Moreover, the proposed combined algorithm achieved an accuracy-improved result even in the dead-zone of the conventional algorithm, as shown in Figure 13c. The average difference of the combined algorithm with a target distance from 100 to 200 μm was 1.60 μm, which was approximately two times smaller than the spectral fringe algorithm of 3.64 μm. However, the performance of the two proposed algorithms significantly deteriorated when the target distance continuously decreased from 100 μm to 60 μm, as shown in the blue region in Figure 13a. This is because the time-domain pulses in the inverse Fourier transformed results of the spectral interference signal could not be selected out with high accuracy even when using these two improved algorithms when the target distance was too short. Employment of an OSA with higher resolution and a femtosecond laser source with a broader spectral width can be helpful in increasing the performance of the two proposed algorithms when the target distance is smaller than 100 μm. This will be verified in future experiments.

## 4. Conclusions

In this paper, two improved data processing algorithms have been proposed for shortening the dead-zone close to the zero-position of measurement, i.e., the minimum working distance of dispersive interferometry using a femtosecond laser, which is a critical issue in millimeter-order short-range absolute distance measurement. The principles of the proposed algorithms, namely, the spectral fringe algorithm and the combined algorithm, have been derived. Meanwhile, we have clarified the impact of the modulated parameter *A* and the data processing procedure of these proposed algorithms using simulation. An experimental setup has been established to carry out short-range absolute distance measurement using dispersive interferometry. The results of the conventional data processing algorithm and those of the proposed algorithms have been compared. It was verified that using the proposed algorithms shortened the dead-zone by half as compared to using the conventional algorithm. Furthermore, owing to the employment of the wrapped phase information directly, the combined algorithm was demonstrated to have better measurement accuracy than the spectral fringe algorithm. The measurement error when using the algorithm in [47] over a distance of 1.2 mm from the zero-point was about 4 μm, while the measurement error when using the proposed combined algorithm in this paper over a larger distance of 10 mm from the zero-point was about 1.8 μm. The latter was approximately two times smaller than the former, which demonstrates the feasibility of the proposed algorithm.

## Figures and Tables

**Figure 1 sensors-23-04953-f001:**
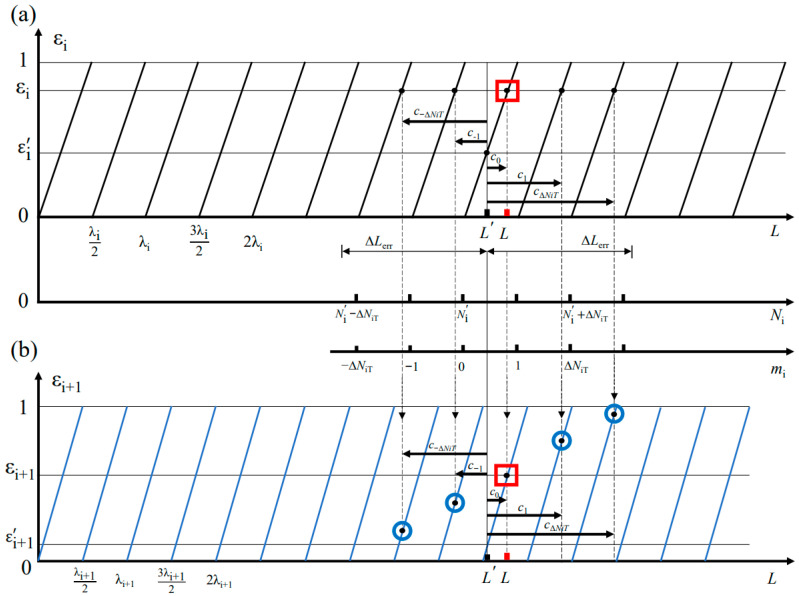
Theoretical analysis of the excess fraction method. (**a**) Excess fraction *ε_i_* of wavelength *λ_i_* periodically varies with increasing target distance *L*, and the other two horizontal axes in the middle represent the acceptable integer part *N_i_* and the acceptable adjustment range for the integer part *m_i_*, respectively. (**b**) Excess fraction value *ε_i+_*_1_ of *λ_i+_*_1_ periodically varies with increasing target distance *L*. The four small blue circles represent the excess fraction *ε*_(*i*+1)*j*_ of $L^{\prime }$ + *c_j_* in wavelength *λ_i+_*_1_, and the magenta square box represents the *ε*_(*i*+1)*j*_, which is the closest to *ε*_(*i*+1)_, whose corresponding *c_J_* is the optimum one.

**Figure 2 sensors-23-04953-f002:**

Interference spectrum with different target distance *L*. (**a**) *L* = 1 mm; (**b**) *L* = 2 mm; (**c**) *L* = 3 mm.

**Figure 3 sensors-23-04953-f003:**
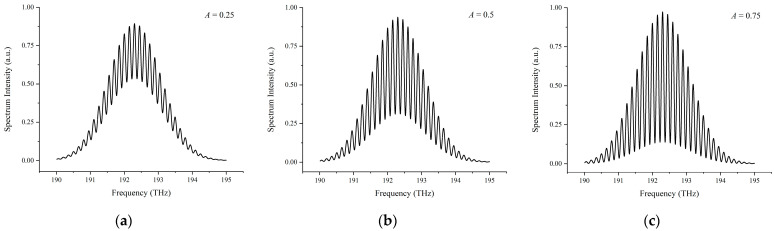
Interference spectrum with different values of modulated parameter *A* for a target distance of *L* = 1 mm. (**a**) *A* = 0.25; (**b**) *A* = 0.5; (**c**) *A* = 0.75.

**Figure 4 sensors-23-04953-f004:**
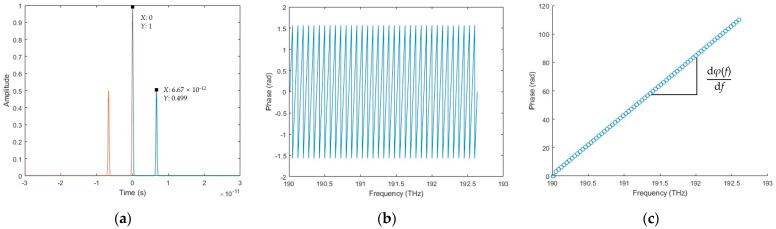
The data processing procedure of the conventional data processing algorithm. (**a**) Inverse Fourier transforming interference spectrum into the time-domain, leading to a rough time delay of 6.67 × 10^−12^ s. The blue and magenta lines represent the results in the positive and negative time-domain, respectively; (**b**) wrapped phase; (**c**) unwrapped phase.

**Figure 5 sensors-23-04953-f005:**
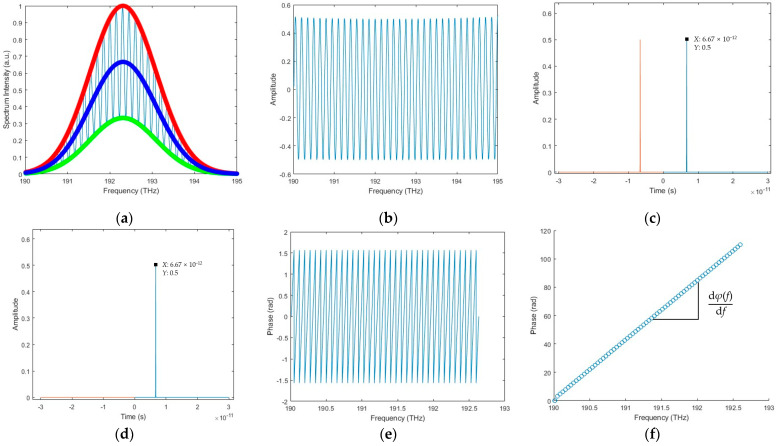
The data processing procedure of the spectral fringe algorithm. (**a**) Extracting the upper and lower envelopes of the interference spectrum, where the red line is the upper envelope, the green line is the lower envelope, and the middle blue line is the spectrum of the source; (**b**) modified spectral interference signal *I_m_*($f_{k}$); (**c**) inverse Fourier transforming and only two sharp peaks can be found, where the blue and magenta lines mean the results in the positive and negative time-domain, respectively; (**d**) time-window filtered peak; (**e**) wrapped phase; (**f**) unwrapped phase.

**Figure 6 sensors-23-04953-f006:**
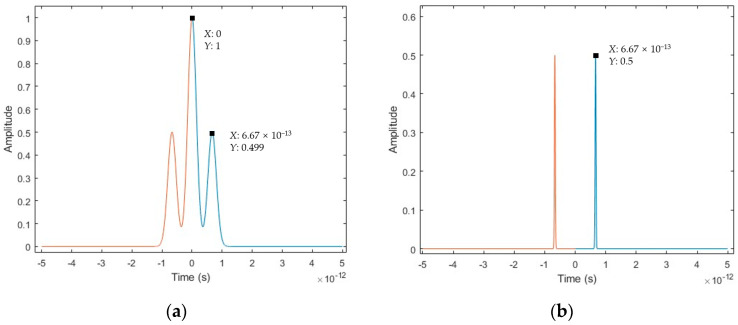
A comparison of the inverse Fourier transform results of the spectral interference signal in the conventional data processing algorithm and the spectral fringe algorithm for a short target distance of 100 μm, where the blue and magenta lines represent the results in the positive and negative time-domain, respectively. (**a**) Inverse Fourier transform results in the conventional data processing algorithm; (**b**) inverse Fourier transform results in the spectral fringe algorithm.

**Figure 7 sensors-23-04953-f007:**
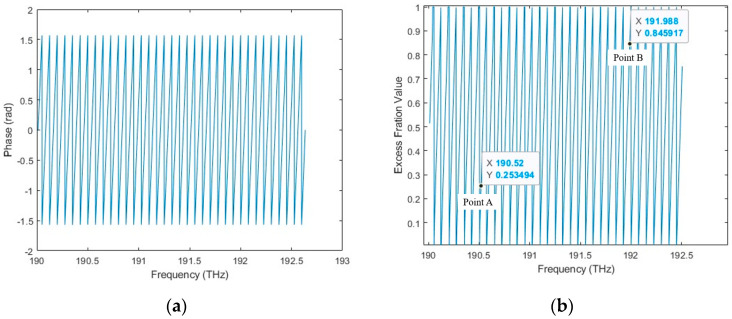
The data processing procedure of the combined algorithm. (**a**) Wrapped phase; (**b**) excess fraction value and two arbitrarily selected Point A (190.519 THz, 0.253494) and Point B (191.988 THz, 0.845917).

**Figure 8 sensors-23-04953-f008:**
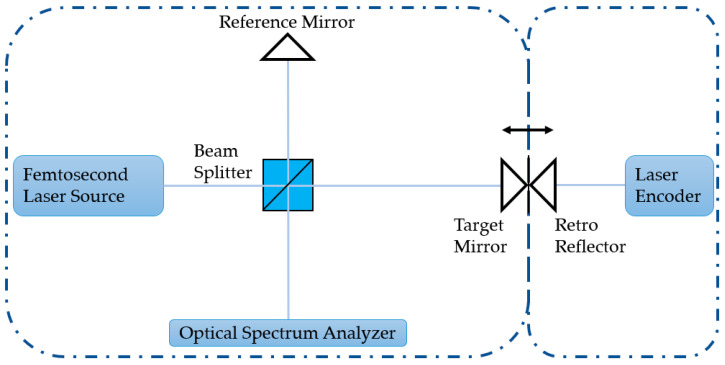
A schematic to show the dispersive interferometry experiment setup.

**Figure 9 sensors-23-04953-f009:**
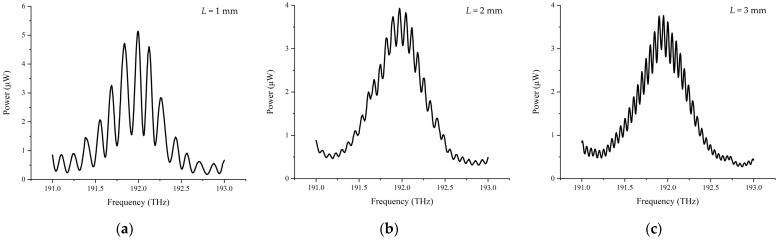
Interference spectrum at different positions. (**a**) The interference spectrum at 1 mm; (**b**) the interference spectrum at 2 mm; (**c**) the interference spectrum at 3 mm.

**Figure 10 sensors-23-04953-f010:**
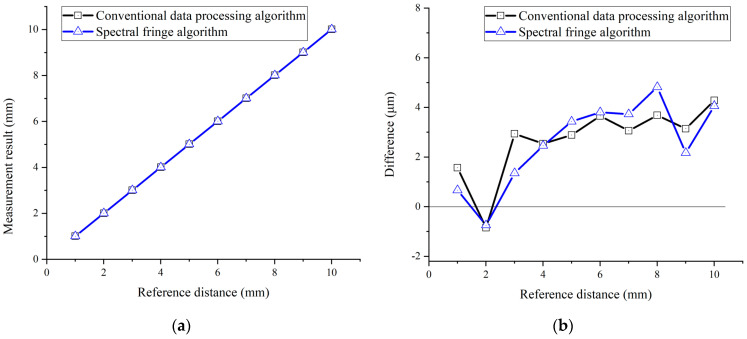
A comparison of the measurement results using the conventional data processing algorithm (black) and the spectral fringe algorithm (blue). (**a**) The measurement results from 1 mm to 10 mm with a step of 1 mm; (**b**) the difference between the measurement results and referenced distance.

**Figure 11 sensors-23-04953-f011:**
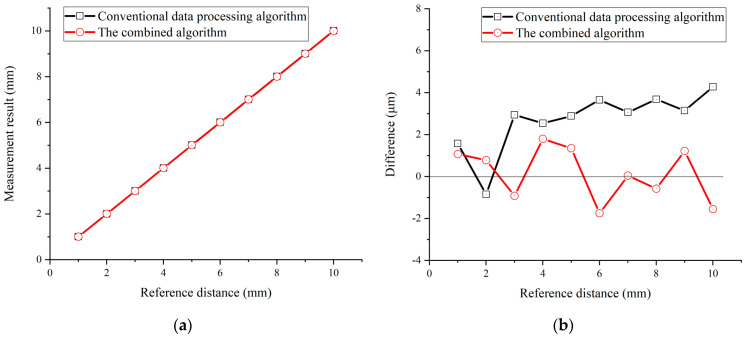
A comparison of the measurement results using the conventional data processing algorithm (black) and the combined algorithm (red). (**a**) The measurement results from 1 mm to 10 mm; (**b**) the difference between the measurement results and referenced distance.

**Figure 12 sensors-23-04953-f012:**
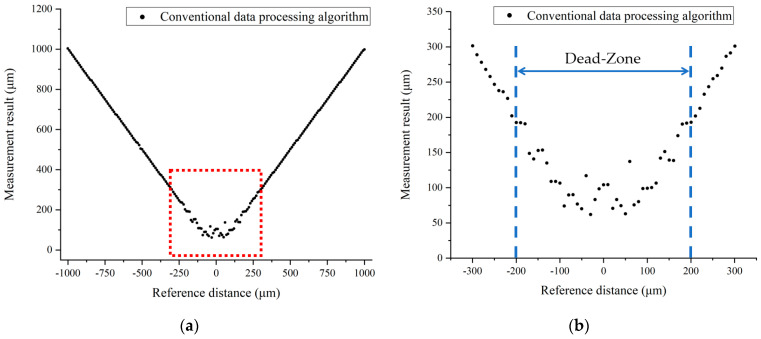
The dead-zone of the conventional data processing algorithm. (**a**) Measurement results of the conventional algorithm from −1 mm to +1 mm; (**b**) amplification of the red box labeled region in the left figure (**a**).

**Figure 13 sensors-23-04953-f013:**
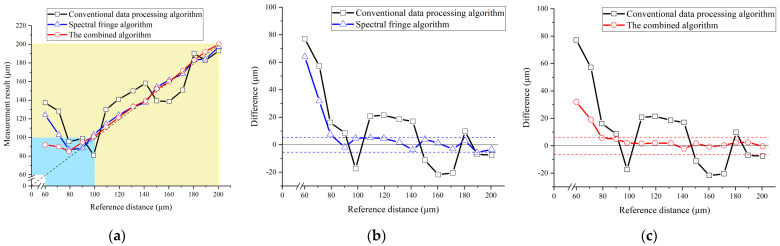
Measurement results of the conventional data processing algorithm, the proposed spectral fringe algorithm, and the proposed combined algorithm from +60 to +200 μm. (**a**) Measurement results of the three algorithms; (**b**) a comparison of the conventional algorithm (black) and the spectral fringe algorithm (blue); (**c**) a comparison of the conventional algorithm (black) and the combined algorithm (red).

## Data Availability

The data presented in this study are available on request from the corresponding authors.

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
