# Peer review of "Improved Algorithms of Data Processing for Dispersive Interferometry Using a Femtosecond Laser"

_sensors, 2023, doi:10.3390/s23104953_

Round 1

Reviewer 1 Report

In the manuscript ‘Improved algorithms of data processing for dispersive interferometry using optical frequency comb’, Liu et al proposed two algorithms in data processing in distance measurement. They provide experimental data to support the advantage of their algorithm as well. The whole manuscript is well-organized. I have following minor comments before the manuscript is published:

1. Can the authors give a brief performance comparison in distance measurement results between the algorithm in [47] and their proposed algorithm? They may add these into the conclusion part or introduction part.

2. I don’t think it is reasonable to use ‘optical frequency comb’ to describe their laser source because there is no frequency stabilization in the mode-locked lasers.

3. The simulation does not show the performance, for example, the accuracy of distance measurement, thus it can not emphasize the advantage of proposed algorithm. Is it possible to provide this simulation?

4. Fig. 8, why the modulated parameter slightly changes with L?

5. In Line 334, it says ‘…because of the stretched pulse interval in the time domain’. I couldn’t understand this statement. Can they give more explanation on this point?

Reviewer 2 Report

Optical frequency comb is very important for precision measurement. The authors proposed a combined algorithm method to shorten the unmeasurable dead-zone, where the narrow linewidth and large distance of pulses benifited the measurement accuracy. The manuscript is organized well and may have potential application in the field of absolute distance measurement. While I have some question as follows.

1. The authors defined the time delay \tao=2n(f)L/c, and then give the expression of the intensity of the corresponding interference signal. How to obtain the expression of the intensity I^k(t) with E_M and E_R? What is physical significance of every term?

2. Line 110, a Gaussian like beam was used to perform Michal interfernce for ultra-narrow linewidth of spectrum. Did the beam size affect the interference effect and the measurement accuracy?

3. Line 233, why a Fabry-Perot eltalon was used to trim down the incident combs' density? Did it affect the linewidth and distance of spectrum in the proposed scheme?

the quality of English is good.

Reviewer 3 Report

Authors report novel algorithms for reducing the dead zone in dispersive optical interferometry measurements. A general overview of the different methods is clearly introduced,correctly written,  thus also including an introduction to the proposed algorithms: spectral fringe algorithm- and the combined algorithm. Both are well introduced with a description of the concerned technics. I outline the following points and questions on the present form : 

- In fig 12 b there is rather large difference between conventional and  combined algorithms, reference distances <100µm. How to interpret this gap-Accuracy of the measurements ?

-Give some characteristics of the fibre laser : pulse width- spectral bandwith - energy per pulse - frequency stability.  

- In Fig11 there is a significant noise contribution in the measurements. Which origin of this noise and impact on the measurement results. 

- Also is it possible to draw comparaisons, analysis and conclusions in term of SNR at the signal detections versus the two algorithms. 

To conclude the manuscript contributes to improve the accuracy of measurements in spectral  interferometry. The methods are correctly developed both for simulations and experimental results . The paper can be considered for publication after taking account of the comments in the final revised form of the manuscript. 

Reviewer 4 Report

This research describes the algorithms for dispersive interferometry data processing. The concepts described in the article are clear, and have good soundness. However, before its publication, the following issues need to be clarified.

Major Issues:

1.       Regarding Proposed Algorithm 1, which is the spectral fringe algorithm, it appears that the uncertainty of the method in Ref [47] is 2.1 x 10^-5, which is better than the proposed algorithm. Therefore, we kindly request the authors to provide additional information to clarify the contribution for the proposed method.

2.       In Figure 12(b), it can be observed that the combined algorithm lacks data in the region of short reference distance. This region coincides with the dead zone of the conventional algorithm, but the absence of data in the same region renders it difficult to compare the performance of the two algorithms.

Other Issues:

1.       May I kindly request the authors to provide the same graphical representation as in Figure 11 but with the proposed algorithm? This would enhance readers' understanding of the research's contribution.

2.       Could author provide the specification of the proposed experiment system? For a measurement system, the measuring range, resolution, uncertainty, and sampling rate. This may help readers to better evaluate the proposed algorithm and system.

Round 2

Reviewer 3 Report

In the revised manuscript the authors have welll considered all the comments of the review. Clear analysis on several questions have been correctly developed thus leading to a valuable increase of the quality of the paper. This revised form is now well suited for publication in the journal. 

Reviewer 4 Report

All the previous questions had been replied to, and I have no further questions for the revised manuscript. As a result, I recommend it for publication.